# Comparative Study on the Joint Biomechanics of Different Skill Level Practitioners in Chen-Style Tai Chi Punching

**DOI:** 10.3390/ijerph19105915

**Published:** 2022-05-13

**Authors:** Hongguang Hua, Dong Zhu, Yifan Wang

**Affiliations:** 1Department of Physical Education, Hangzhou Dianzi University, Hangzhou 310018, China; huahongguang@hdu.edu.cn; 2School of International Education, Shanghai University of Sport, Shanghai 200438, China; 3Department of Physical Education, Zhejiang University, Hangzhou 310058, China; zjuwangyifan@zju.edu.cn

**Keywords:** Tai Chi, Chen-style, punch, biomechanics

## Abstract

Purpose: “Yan Shou Hong Cui” is a representative punch in Chen-style Tai Chi. The training is an important factor in affecting the effect of Tai Chi practice. Joint angles are the most intuitive way to evaluate motion. The purpose of this study is to compare the movements of Tai Chi masters and beginners’ movements through the analysis of joint angle and punching effect, and explore the influence of training years on the power generation of Tai Chi punches. Methods: There were 14 Chinese Chen-style Tai Chi subjects recruited for this study. They were divided into the master group (*n* = 7, age of 38.14 ± 10.42 years, height of 1.68 ± 0.06 m, weight of 71.33 ± 8.61 kg) and the beginner group (*n* = 7, age of 38.00 ± 11.94 years, height of 1.69 ± 0.07 m, weight of 70.14 ± 9.79 kg). The typical movement from Chen-style Tai Chi is called Yan Shou Hong Cui. All subjects were asked to perform the action three times, and the way of force was applied each time. The kinematic and kinetic characteristics of Tai Chi were analyzed by the VICON 3D motion analysis system (100 Hz) with 16 MX13 cameras, and the punch efficiency was measured by the Chinese Kung Fu Dummy (80,000 Hz). Results: The experimental results show that the shoulder, elbow, and hand movements of the master group are more precise and the force strength is more vigorous (master group: the peak angular velocity of the shoulder joint was −324.12 ± 50.88°/s, the angular velocity of the elbow joint was −112.83 ± 56.98°/s, and the hand angular velocity was −121.69 ± 49.55°/s; beginner group: shoulder angle velocity was −281.17 ± 30.56°/s, the elbow angle speed was −263.64 ± 68.63°/s, and the hand angle speed was −36.80 ± 12.53°/s). The rotation angle and rotation speed of the hip joint in the master group were significantly higher than those in the beginner group (as for the hip rotation angle, the master group was 64°, which was significantly higher than the beginner group’s 44°, and there was a significant difference (*p* = 0.019 < 0.05)). In terms of rotation speed, the peak value of the master group was 370.3 ± 94.8°/s, which was significantly faster than that of the beginner group at 210.4 ± 56.1°/s, and there was a very significant difference (*p* = 0.003 < 0.01). The master group’s punch effect acceleration (496.39 ± 256.52 m/s^2^) was significantly higher than that of the beginner group (396.90 ± 116.79 m/s^2^). Conclusion: People who practice Tai Chi for a long time differ from the beginners in terms of joint flexibility, punching posture, and the effect of application.

## 1. Introduction

Traditional Chinese sports were a treasure of the Chinese civilization. It embodies the broad wisdom of the Chinese for thousands of years. Exercise therapies, such as Wu Qin Xi, Ba Duan Jin, and Tai Chi, have continued to grow in popularity and have been proven to have unique curative effects to the world. Studies have shown that Tai Chi has positive effects on cardiovascular, respiratory, and immune functions [1,2,3,4]. In addition, Tai Chi can also increase the range of motion, strengthen the muscle strength, and improve the balance of the human body [5,6,7]. In conclusion, Tai Chi has a positive impact on human health, and it is of great significance to conduct in-depth research on it [8,9]. However, few scholars have performed quantitative research on the specific movements of Tai Chi and seldom discuss its internal force mechanism. Tai Chi has been full of mystery for a long time due to the lack of quantitative research on the content and mechanisms [10,11]. It is generally believed that using Tai Chi to beat people can suggest that “one touch is flying” and “the other is wounded” [12]. If Tai Chi and even Chinese martial arts really want to be extended into the world, they must be scientific, quantitative, and standardized [13]. Therefore, it is very necessary to scientifically and accurately evaluate Tai Chi and explore its internal mechanism.

Chen-style Tai Chi is the typical representative of the traditional Chinese Kung Fu. “Yan Shou Hong Cui ′′ is one of the typical and important movements in Chen-style Tai Chi, which contains all its basic movements. Therefore, this study chooses to study this representative action. “Yan Shou Hong Cui” is a complex technical movement, including hand punching, waist twisting, leg extension, center of gravity movement, and hip and knee twisting. It is a full-body movement that requires the movement of several major joints of the body, including the shoulder, elbow, wrist, hip, and knee and ankle joints. The movement of the hip and knee joints is the core of the movement to generate power. Therefore, this study focused on the activity of each joint.

On the other hand, we suspect that years of training may affect Tai Chi movement norms and power generation methods [14,15]. In addition to the basic movements, the traditional teaching of Tai Chi focuses more on the application of inner Qi, which can be understood as a complex neuromuscular control. This process is difficult to describe in words and requires learners to understand it by themselves. Therefore, Tai Chi masters generally spend many years to practice in order to achieve a high level, mainly in the use of Qi and the process of generating power. Based on the content above, the hypotheses of this study are: 1. The punch effect by Tai Chi masters is superior to that of Tai Chi beginners; 2. The postures and power release skills are different between the Tai Chi masters and beginners due to the differences of punch efficiency. The purpose of this study is to design a case–control experiment to investigate the Tai Chi punching power mechanism and explore the influence of training years on the power generation of Tai Chi punches by comparing the punching differences of Tai Chi exercisers in different training years.

## 2. Methods

Fourteen subjects were eventually enrolled, depending on their training level. First, we randomly selected 7 volunteers of the 50 Tai Chi masters from all over China as the master group (age of 38.14 ± 10.42 years, height of 1.68 ± 0.06 m, and weight of 71.33 ± 8.61 kg); the 50 Tai Chi masters were recommended by the general secretary of the Chen-style Tai Chi Association because their Tai Chi skill presented a high level of Chen-style Tai Chi. Then, we selected seven beginner Tai Chi practitioners based on their age and figure of the master group as a beginner group (age of 38.00 ± 11.94 years, height of 1.69 ± 0.07 m, and weight of 70.14 ± 9.79 kg). In order to control the irrelevant variables (age, height, and weight), the subjects in the beginner group were recruited based on the age, height, and weight that were matched to the master group; the error was controlled within the height of ±3 cm, weight of ±3 kg, and the age of ±3 years (Table 1 and Table 2). Using a one-way ANOVA to test homogeneity, we set *p* > 0.05. A typical movement in Chen-style Tai Chi called cover hands and punch with arm (Yan Shou Hong Cui) was selected as the experiment’s object (Figure 1). 

### 2.1. Variables and Experimental Equipments 

Joint angle and angular velocity are the most intuitive ways to describe movements, so this study also focused on them. This study used the British Vicon 3D Motion Analysis System with 16 MX13 cameras (100 f/s) for motion capture to calculate the joint angles and angular velocities.

To reflect the impact of a punch, we evaluated it using the acceleration of the object being hit. The Chinese Kung Fu Dummy (jointly developed by Shanghai University of Sport and Chengdu Fang Tuo Dummy Company, 80,000 Hz) was used to evaluate the impact of punches and high-speed cameras (Redlake MASD, Tucson, AZ, USA, MotionPro X-4, 500 f/s) were used to record the time of the fist making contact on the dummy. The dummy was a key device in this study, so it is necessary to introduce it in detail. The dummy was designed for the experiment to test the effects of the Chinese martial arts master punching on the human body. It presented the basic characteristics of Chinese adults, 50 percent with 9 accelerometers in dummy. Its height was 170 cm and its weight was 68.4 kg. Piezoelectric X, Y, and Z three-axis acceleration composite sensors were installed on the head. The acceleration changes in three directions could then be collected. Other accelerometers were mainly installed in the dummy’s body, which were placed in 6 different positions: sternum handle, heart, liver, left lung, right lung and stomach; the sensors collected the specific acceleration of the impact of the dummy’s internal organs. We marked the hit position with a red brush to calibrate the dummy sternum handle before the experiment. During the experiment, we used a high-speed camera to monitor and record the test process to ensure the identity and accuracy of the hit position (Figure 2).

### 2.2. Experimental Procedure

Before the experiment began, the subject performed warm-up preparation activities. At the same time, the experimenters started to calibrate the experimental equipment. The reflective mark balls were pasted to perform VICON static calibration and complete static portrait data collection. The Kistler 3D Force Test Stations device was calibrated to ensure the dynamometer data were correct. High-speed monitoring was used to ensure the accuracy of the hitting position and check the validity of the four-channel acceleration of the Chinese Kung Fu dummy’s sternal notch. Then, the participant was informed of the experimental process and the requirements that needed the participant to complete at least three punches successfully. Each administrator of equipment examined the authenticity and validity of the data individually. To avoid lack of data samples in post processing (Figure 3), each action was required to be successful at least 3 times. As shown in (Figure 4), the beginner group followed the same procedure.

### 2.3. Statistical Analysis

After data collection, we used the Visual3d Version 3.34.0 software (Crunchbase Inc., London, UK)to construct a 3D skeleton model of the human body, where the data were used as the low pass filter with 9 Hz (Figure 5). Additionally, we calculated the joint angular velocity (°/s), hip rotation angle (°), and angular velocity (°/s). The Kung Fu dummy data were imported into the Yep software, which was provided by Jiang Su sensor Co. Ltd. (Yangzhou, China) to process data for the changes in the acceleration (m/s^2^) of each viscera during the beating process. All experimental data were calculated based on Spss17.0 Statistics Software (SPSS Inc., Chicago, IL, USA), the paired sample *t*-test, and correlations were chosen to analyze the data. In addition, we also used Origin7.0 (OriginLab Corporation, Northampton, MA, USA) drawing and other software for the time standardization of graphic processing.

## 3. Results

### 3.1. Speed of Hand, Elbow, and Shoulder Angle

By comparing the peak values of the master group (Figure 6 and Figure 7), the peak angular velocity of the shoulder joint was −324.12 ± 50.88°/s, the angular velocity of the elbow joint was −112.83 ± 56.98°/s, and the hand angular velocity was −121.69 ± 49.55°/s. The peak values of the three joints in the beginner group were shoulder angle velocity −281.17 ± 30.56°/s, elbow angle speed −263.64 ± 68.63°/s, and hand angle speed −36.80 ± 12.53°/s. Besides, the elbow speed of the master group was slower than that of the beginner group, and the shoulder angular velocity and the hand angle speed were faster than those of the beginner group. From the angle of every joint velocity, this paper found that the trajectories of the joints in the master group begin to reach the maximum peak repeatedly. In addition to the fluctuation of the elbow joint, the angle velocity locus of hand and shoulder was relatively flat in the beginner group. The order of subjects to reach the maximum peak of the joint angular velocity in the two groups was different. The master group was in the order of hand (41%)–elbow (45%)–shoulder (55%), while the beginner group was in the order of shoulder (42%)–elbow (44%)–hand (46%).

### 3.2. Hip Rotation Angle

The data show that the hip joint of the master group has a large pre-swing movement at the end of the energy storage, and the angle of the hip joint was about −42.3 ± 8.6° from the rotation angle of the hip joint (Figure 8). At the beginning, the angle of the hip joint began to rise to 21.7 ± 6.3°, and the rotation angle of the hip joint was 64°, while the beginner group first changed very slow by −18.3 ± 5.2°, and then steadily increased to 25.7 ± 4.6°. The rotation angle of the hip joint was 44°. The rotation angle of the hip joint in the master and beginner groups were statistically significant by paired sample *t*-test (*p* = 0.019 < 0.05). 

### 3.3. Hip Angular Velocity

Angular velocity is a key index to measure the speed of a movement. The comparison of the angular velocity between the two groups showed (Figure 9) that the peak angular velocity of the beginner group appeared earlier than the master group, and the angular velocity of the master group changed more significantly. It rapidly declined from the initial speed of 0°/s to the peak value of 370.3 ± 94.8°/s, forming a steep inverted “U” type. In contrast, the trend of angular velocity in the beginner group was relatively stable. The peak value appeared at 210.4 ± 56.1°/s, decreasing slowly. The paired samples *t*-test showed a significant difference between the two groups in the maximum angular velocity (*p* = 0.003 < 0.01). In addition, the angular velocity and rotation angle of the hip joint showed that the master group had a linear relationship between the angular velocity of the hip joint and the angle of rotation (R^2^ = 0.727), but in the beginner group it was not obvious (R^2^ = 0.417) (Figure 10).

### 3.4. The Chinese Kung Fu Dummy Result

The acceleration in the master group was significantly higher than that in the beginner group by comparing the two groups’ hitting effects (Table 3). Regarding the beating position (sternum handle), the master group (496.39 ± 256.52 m/s^2^) beat higher than the beginner group (396.90 ± 116.79 m/s^2^). The acceleration of the heart was 92.14 ± 57.44 m/s^2^ in the master group, while it was 39.19 ± 8.39 m/s^2^ in the beginner group. The liver position in the master group was 256.09 ± 232.60 m/s^2^, while in the beginner group it was 118.46 ± 69.94 m/s^2^. The left lung part of the master group was 162.96 ± 134.25 m/s^2^, while in the beginner group it was 68.23 ± 32.83 m/s^2^. The right lung area in the master group was 117.87 ± 86.60 m/s^2^, while in the beginner group it was 59.42 ± 12.93 m/s^2^. The stomach area in the master group was 225.94 ± 176.04 m/s^2^, while in the beginner group it was 97.04 ± 43.70 m/s^2^.

## 4. Discussion

This study analyzed the joint angle, angular velocity, and acceleration of the hit object, and discussed the differences between Tai Chi masters and beginner people regarding their punch. Previous studies have focused on boxing, karate, and taekwondo [16,17,18,19]. Schwartz et al. [16] compared karate and boxing dummies and found that the gloves and pads used to protect or cushion the dummy had no significant difference in acceleration over different acceleration ranges. Some researchers compared “three-inch fist” and “counter-fist” by using a three-dimensional force measuring table and a three-dimensional kinematics analysis system [19]. The study found that the maximum force and force attenuation rate of a “counter-punch” was significantly better than that of a “three-inch punch”, and there was no significant difference in the impact force on the target. The study found that the “three-inch fist” had a slight advantage in situations where the opponent was at a short distance or with the tactical goal of changing the opponent’s center of gravity.

The basic movement of a human’s upper limbs has three main forms of pushing tension and whipping pushing is the process of the upper limbs’ extensors to overcome resistance and the joint changes from flexion to extension, such as shot putting, weightlifting, and pushing barbell. The right boxer that covers the hand was also consistent with the form of pushing. The results show that the range of motion of all joints did not reach the end of the range of motion during the whole movement, and even the elbow did not reach a straight state. Tai Chi requires the body to be at the best angle of force before exerting force. When the force is applied to the object, the joint should not be at the end of the range of motion, but should be in a slightly flexed state, so that the force is transmitted to the target through the bones and joints. Experienced Tai Chi masters all know that if the elbow joint is extended while exerting force, it is easy to be damaged by the reaction force after acting on the object, and it is easier to exert force on the hand by bending slightly.

Tai Chi Master Xin Chen said that the elbow should be driven by hand, the shoulder by elbow, the lower knee by the feet, and the thigh by the knee [20]. From the sequence of the maximum angular velocity of the shoulder elbow joint angle in the master group, the angular velocity of the hand reached the maximum peak at about 41% of the time, the elbow joint followed by about 45%, and the shoulder joint reached the maximum peak of about 55%. This argument was similar to handing up the elbow and following the shoulder. The order of activity of large muscle groups driving small muscle groups was just the opposite. The large joints with whipping movements drive small joints.

The main characteristic of brachial beating by covering the hand was shaking strength, such as when Tai Chi Master Zhao Kui Chen used figurative metaphors for it. He said it is as the energy of the walking cattle shaking off the dust after they rolled in the land, throwing the fist out when you punch and shaking your arm out when emitting arm force. Loosening the power of a projectile and shaking requires a relaxed and natural situation. In the force moment, it is as Mars burning the skin, suddenly surprised, cold out, simply powerful. Additionally, in actual combat, it also requires the whole body to be as an electric shock, as if Mars burns suddenly, and the so-called brainstorm cannot escape. From the angular velocity trajectory of the shoulder–elbow–hand joints, the moment of the master group’s instantaneous momentum was undulating, as a spring-like return, and finally reached a point, which was completely in line with the distribution of the force. While the beginner group could not reflect the dynamic characteristics and the ending action without dropping strength, dropping strength issued a focal point to body stress, and was seemingly harmless but flashy without substance.

Western scholars have shown that, when a certain action is completed, the conscious backswing of the body can cause the tense contraction of the relevant muscles and tissues [4,5,7]. The muscles were pre-loaded and stored in a certain amount of elastic energy for forwarding swing. This view coincides with the essentials of Tai Chi energy storage. Some scholars think that the strength of Tai Chi is force with storage [21]. Strength storage is the prerequisite and necessary condition for strength, and strength is the ultimate goal and inevitable result of storing strength [20]. The boxing theory explains that strength storage is as opening a bow and forcing strength as an arrow. Therefore, forcing strength depends on strength storage [22]. Force strength can achieve the characteristic of galloping the world to the core by strength storage. By comparing the hip angle (Figure 8), it was found that the hip joint angle swung back to −42.3 ± 8.6° when the master group ended. In contrast, the swung amplitude of the beginner group was only −18.3 ± 5.2°, and the difference between the two groups was about 24°.

The amplitude of the rear swing of the master group was more than one time that of the beginner group. The paired sample *t*-test (*p* = 0.019 < 0.05) showed the significant difference. It showed that the master group has more strength with more strength storage. Liking the strength of drawing the bow, the more perfect the bow is, the greater the tension. The larger amplitude of the swing provides a greater space and momentum for further raising the rotational angular velocity. From the master group, the angular velocity of the hip joint (Figure 9) rapidly increased from the initial velocity of 0°/s to the maximum value of 370.3 ± 94.8°/s, which proved this point forcefully. In contrast, the beginner group was not sufficiently vigorous and smooth due to the small amplitude of the pre-swing and insufficient storage strength. When the strength was strong, the trend of hip angular velocity was relatively gentle with a maximum value of 210.4 ± 56.1°/s. There was a very significant difference between the master and beginner groups (*p* = 0.003 < 0.01). As Xiao-wang Chen taking opinions of punch with covering hands, the front elbow was symmetrical, the waist was as the axle, the air was as a wheel, and the rotation force of the crotch waist was fully used. Therefore, there was the incisive argument of power releasing in the crotch, which has its own spring force.

The limitations of this study mainly include the following two points: one is that the research objects were only males, and the other is that the force was not analyzed. The reason for recruiting only men is that, in ancient China, there was a tradition that only men were taught Tai Chi. In modern times, even though Tai Chi has gained popularity, the proportion of men is still higher than that of women. In addition, Tai Chi masters were generally male. So, this study only selected men as research objects. Furthermore, it is a pity that the force of the punch was not accurately measured. In the future, women can be used as research objects and the effect of force can be discussed.

## 5. Conclusions

The study found that four joints in the master group had a better coordination than those in the beginner group by comparing the angular velocity between the four shoulder, elbow, hand, and hip joints. Moreover, the ups and downs of sports lines reflect the shakes and strength in Tai Chi and the placement was relatively clear. The beginner group did not have a shaking re-entry movement route of the strong force, strong impact force was out of order, and penetration was not strong. Thus, people who have been practicing Tai Chi for a long time differ from the novices in terms of joint flexibility, punching posture and applying force, resulting in a different punching efficiency. Therefore, the beginners should pay more attention to the coordination between the upper limb shoulder, elbow, and wrist joints, as well as the lower limb hip and knee joints on the transmission of force. In addition, the joint angular velocity can be increased through explosive-force exercises to achieve a faster punching speed.

## Figures and Tables

**Figure 1 ijerph-19-05915-f001:**
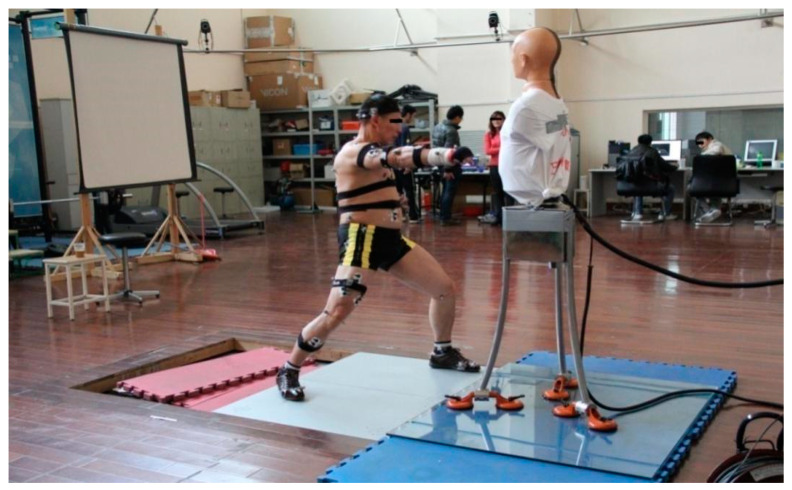
Cover hands and punch with arm action schematic diagram.

**Figure 2 ijerph-19-05915-f002:**
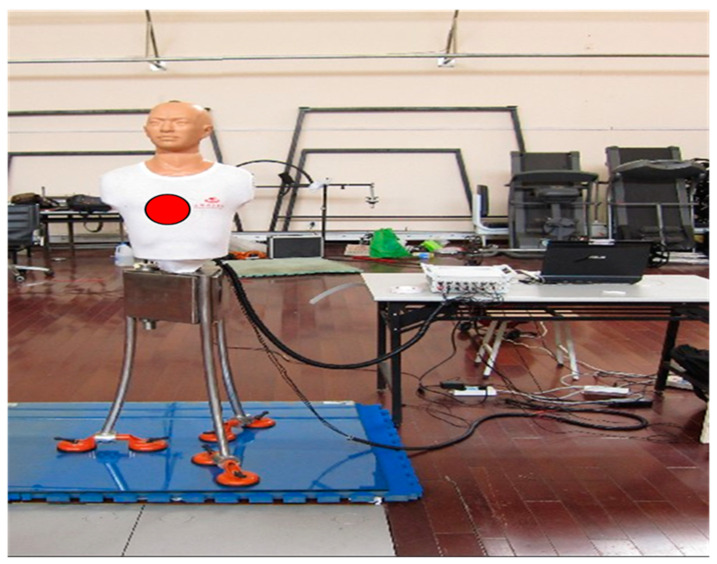
Hit position of the Chinese Kung Fu dummy.

**Figure 3 ijerph-19-05915-f003:**
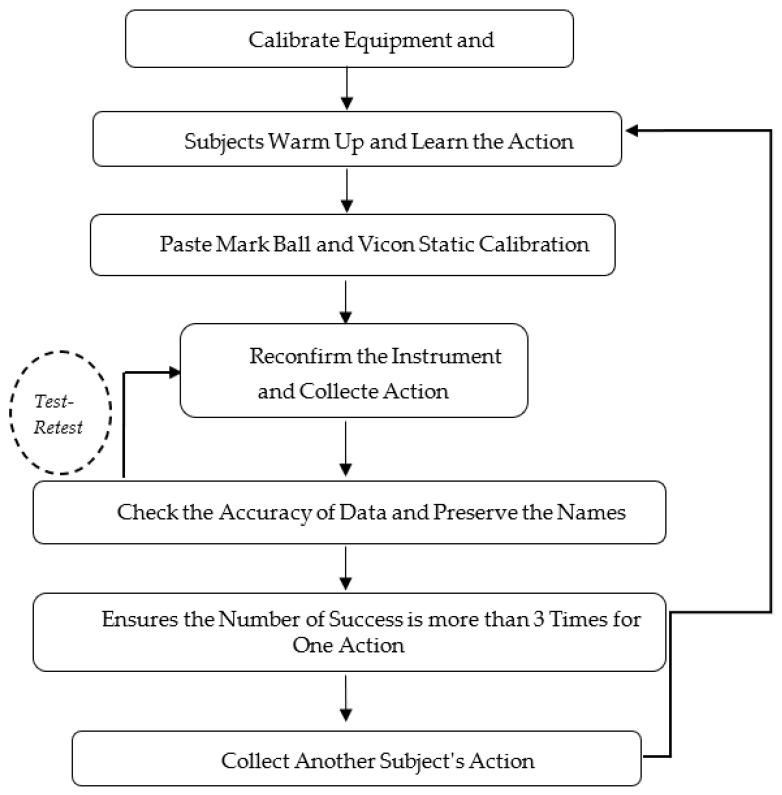
Process of the action test.

**Figure 4 ijerph-19-05915-f004:**
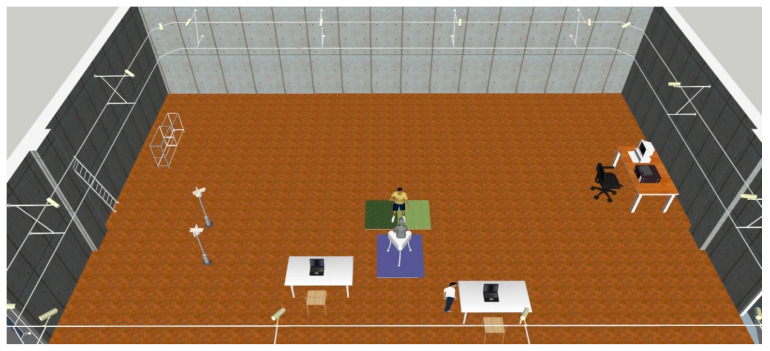
The 3D renderings of the experimental site and equipment installation location.

**Figure 5 ijerph-19-05915-f005:**
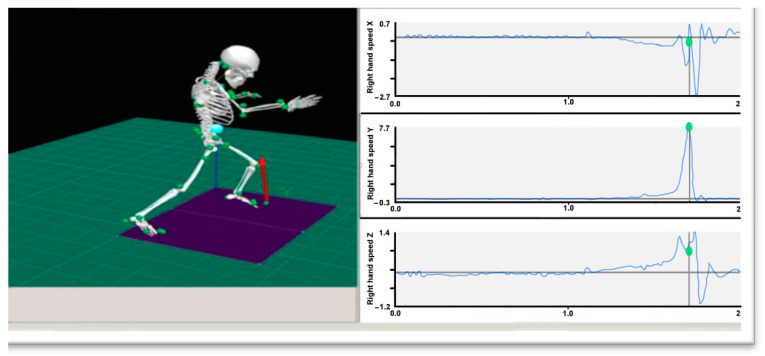
Vicon 3DMotion Resolution Interface diagram.

**Figure 6 ijerph-19-05915-f006:**
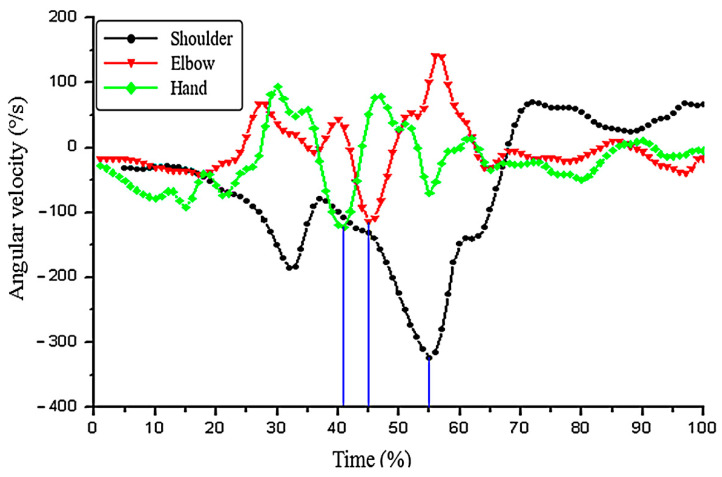
Angular velocity change in the shoulder–elbow–hand joints in the master group (*n* = 7).

**Figure 7 ijerph-19-05915-f007:**
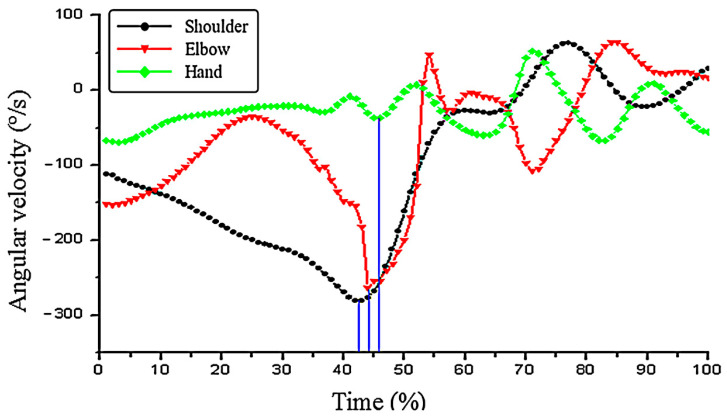
Angular velocity change in the shoulder–elbow–hand joints in the beginner group.

**Figure 8 ijerph-19-05915-f008:**
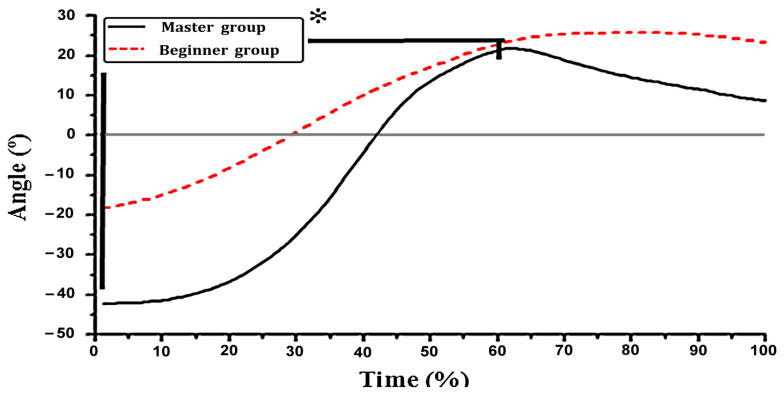
Change in the hip joint rotation angle. * indicates *p* < 0.05, ** indicates *p* < 0.01.

**Figure 9 ijerph-19-05915-f009:**
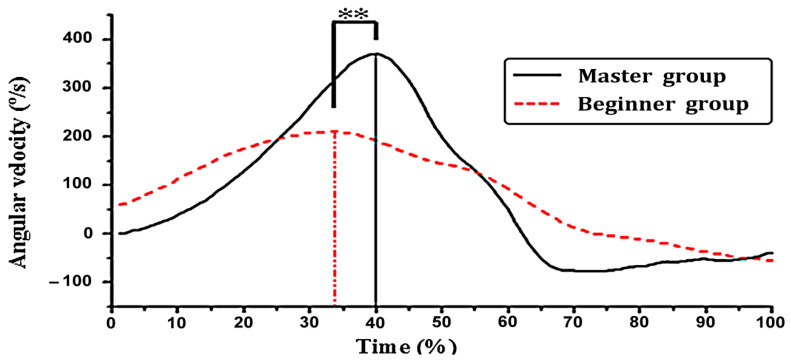
Change in hip joint angular velocity. * indicates *p* < 0.05, ** indicates *p* < 0.01.

**Figure 10 ijerph-19-05915-f010:**
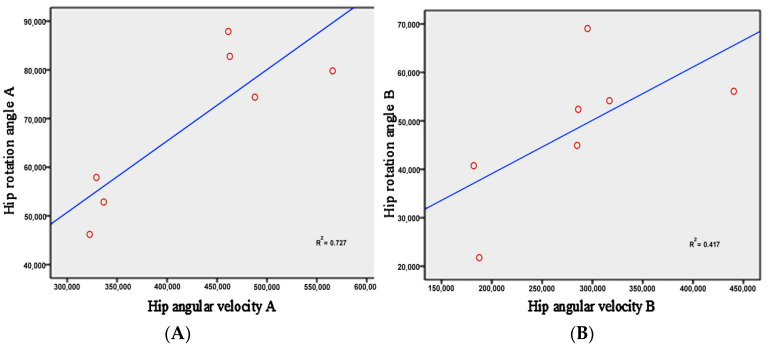
Correlation comparison of the hip angular velocity and hip rotation angle. (**A**) is the master group; (**B**) is the beginner group.

**Table 1 ijerph-19-05915-t001:** Subjects’ basic information.

Variables	Master (*n* = 7)	Beginner (*n* = 7)	*p*
Height (m)	1.68 ± 0.06	1.69 ± 0.07	0.687
Weight (kg)	71.33 ± 8.61	70.14 ± 9.79	0.814
Age (years)	38.14 ± 10.42	38.00 ± 11.94	0.981
BMI (kg/m^2^)	25.28 ± 2.20	24.40 ± 1.99	0.444
Experience (years)	22.57 ± 5.41	0.500 ± 0.18	0.002 *

Note: *p* is the significance of the one-way ANOVA; * indicates that there is a significant difference.

**Table 2 ijerph-19-05915-t002:** Subjects’ detailed information.

Groups	Name	Height (m)	Weight (kg)	BMI	Age (years)	Experience (years)
Master group	Chen XX	1.75	70.0	22.86	54	30
	Xiong XX	1.63	64.3	24.20	28	18
	Du XX	1.61	68.7	26.50	47	28
	Shi XX	1.70	65.3	22.60	24	15
	Chen XX	1.65	70.0	25.71	35	20
	Chen X	1.77	90.0	28.72	38	22
	Xie XX	1.64	71.0	26.40	41	25
Beginner group	Gong X	1.77	70.0	22.34	56	0.4
	Ji XX	1.67	65.0	23.30	27	0.8
	Gao X	1.64	65.0	24.17	50	0.3
	Tang XX	1.69	66.0	23.11	22	0.6
	Xu XX	1.66	67.0	24.31	35	0.3
	Su XX	1.80	92.0	28.40	37	0.5
	Sun XX	1.62	66.0	25.15	39	0.6

**Table 3 ijerph-19-05915-t003:** Comparative analysis of the acceleration of hitting the dummy (unit: m/s^2^; A is the master group; B is the beginner group).

	N	Average Value	Standard Differences	Standard Error of Average Value	*p*
Head X axis A	7	261.89	170.08	64.28	0.075
Head X axis B	7	125.26	50.81	19.20	
Head Y axis A	7	595.10	849.85	321.21	0.066
Head Y axis B	7	99.12	45.22	17.09	
Head Z axis A	7	324.17	379.18	143.32	0.082
Head Z axis B	7	105.02	45.21	17.09	
Sternal handle A	7	496.39	256.52	96.96	0.011 *
Sternal handle B	7	396.90	116.79	44.14	
Cardiac A	7	92.14	57.44	21.71	0.004 *
Cardiac B	7	39.19	8.39	3.17	
Liver A	7	256.09	232.60	87.91	0.022 *
Liver B	7	118.46	69.94	26.44	
Left lung A	7	162.96	134.25	50.74	0.006 *
Left lung B	7	68.23	32.83	12.41	
Right lung A	7	117.87	86.60	32.73	0.006 *
Right lung B	7	59.42	12.93	4.89	
Gastric A	7	225.94	176.04	66.54	0.008 *
Gastric B	7	97.04	43.70	16.52	

Note: *p* is the significance of the paired sample *t*-test; * indicates that there is a significant difference.

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
