# Peer review of "Comparative Study on the Joint Biomechanics of Different Skill Level Practitioners in Chen-Style Tai Chi Punching"

_ijerph, 2022, doi:10.3390/ijerph19105915_

Round 1

Reviewer 1 Report

Introduction

While this section is pretty clear I miss a more in depth description of what are punches and what are the main postures and skills required in Tai Chi.

How normal people become a Tai Chi master? What are the skills and the steps that lead to mastery level in this modality?

Methods

My intuition tells me that only men were recruited for this study. Please, if this is the case clarify and explain why

Years of Tai Chi should be shown in both groups

I do not agree that height, age and weight are irrelevant, since they are variables that influence muscular strength, power and ability.

It is somehow surprising that by randomly selecting Tai Chi beginners, mean age of both groups are almost identical. Was not age an inclusion criteria for being included in this group?

As previously stated, there is a need to identify the variables linked to the punch performance and describe how are they going to be measured ( I mean separately and not as a whole as it is shown in the actual version of the manuscript).

Results

Good that variables are explained separately

Line 189, there is a typo: it was shows

It is not clear what are the authors referring to by liver position, lung position…are they describing the accuracy (hitting target) of the punch?

Why was not  muscular strength (kg or newtons) assessed? Even using a hand-held dynamometer could have been very useful.

Discussion

This section is very narrative and speculative. There is a need to synthetize. The authors should compare and explain differences between the parameters assessed in both groups. It is experience, mastery level, ability…why makes a master different from a beginner when it comes to punching?

Is there normative values of the parameters assessed to compare with? It could have been interesting to test normal people punching and see whether beginners show higher punching skills. If this is not possible, using normative values for comparison is an accurate solution.

Is there values of the parameters assessed obtained from other combat sports? This should be also mentioned.

Limitations should be acknowledged. For instance, the sample was not representative (in the case that women were not included in the research).

Conclusion

While the investigation is original, results are somehow expected. It could be of value adding information regarding how master achieve a higher punching performance. This could show the path for future practitioners whishing to become masters.

Author Response

Dear Reviewer

Thank you for your valuable comments on this article. After careful reading, we have revised the existing problems in the article. The following are explanations of your questions.

1.Introduction

1.1 While this section is pretty clear I miss a more in depth description of what are punches and what are the main postures and skills required in Tai Chi.

A: Relevant content has been added to the original text.

1.2 How normal people become a Tai Chi master? What are the skills and the steps that lead to mastery level in this modality?

A: Relevant content has been added to the original text.

2.Methods

2.1 My intuition tells me that only men were recruited for this study. Please, if this is the case clarify and explain why.

A: The reason for recruiting only men is that in ancient China, there was a tradition that only men were taught Tai Chi. In modem times, even though Tai Chi has gained popularity, the proportion of men is still higher than women. In addition, Tai Chi masters were generally male. So this study only selected men as research objects.

2.2 Years of Tai Chi should be shown in both groups.I do not agree that height, age and weight are irrelevant, since they are variables that influence muscular strength, power and ability.It is somehow surprising that by randomly selecting Tai Chi beginners, mean age of both groups are almost identical. Was not age an inclusion criteria for being included in this group?

A:Sorry for the confusion caused by the language expression. In fact, we agree that height, age and weight are relevant. Because Tai Chi masters are randomly selected from a group recommended by the general secretary of the Chen Style Tai Chi Association, age and size were dispersed. But we take age, height and weight as inclusion criteria when recruiting beginners. In order to exclude the influence of age, height and weight, we specially selected beginners with similar situations according to the age and body type of each Tai Chi master.

2.3 As previously stated, there is a need to identify the variables linked to the punch performance and describe how are they going to be measured ( I mean separately and not as a whole as it is shown in the actual version of the manuscript).

A:It has been modified in the article.

3.Results

3.1 Line 189, there is a typo: it was shows

A:It has been modified in the article.

3.2 It is not clear what are the authors referring to by liver position, lung position…are they describing the accuracy (hitting target) of the punch?

A:The main idea here is that there are multiple accelerometers that were in the different organs of the Chinese Kung Fu Dummy. Tai Chi has obvious differences with other boxing sports in the impact effect. When Tai Chi strikes the human body, the force is transferred to the inside of the body, causing no serious damage to the surface but a wide range of the inner damage to the internal organs. The dummy can collect the data of a punch to estimate the damage to the internal organs.

3.3 Why was not muscular strength (kg or newtons) assessed? Even using a hand-held dynamometer could have been very useful.

 A:The force sensors were installed on the heart, liver, left lung and right lung respectively. However, due to the small force value of the internal organs, the maximum force is not more than 20 newtons, which is very different from the test results of foreign counterparts. We doubt the accuracy of the test results and therefore do not discuss force values in this study.

4.Discussion

4.1 This section is very narrative and speculative. There is a need to synthetize. The authors should compare and explain differences between the parameters assessed in both groups. It is experience, mastery level, ability…why makes a master different from a beginner when it comes to punching?

A:It has been modified in the article.

4.2 Is there normative values of the parameters assessed to compare with? It could have been interesting to test normal people punching and see whether beginners show higher punching skills. If this is not possible, using normative values for comparison is an accurate solution. Is there values of the parameters assessed obtained from other combat sports? This should be also mentioned.

A:There is no normative value comparison because there is no accepted uniform standard in the industry. On the other hand, compared with other boxing classes, the movement pattern of Tai Chi is obviously different, so this study did not make a comparison.

4.3 Limitations should be acknowledged. For instance, the sample was not representative (in the case that women were not included in the research).

A:It has been modified in the article.

  1. Conclusion

5.1 While the investigation is original, results are somehow expected. It could be of value adding information regarding how master achieve a higher punching performance. This could show the path for future practitioners whishing to become masters.

A: Relevant content has been added to the original text.

Reviewer 2 Report

Dear author

Any attempt to quantify Tai Chi is good. The graphics and graphs in the document are very interesting.

However, the text of the document lacks a narrative for a scientific understanding of the reader.

Author Response

Dear Reviewer

Thank you for your valuable comments on this article. After careful reading, we have revised the existing problems in the article. The following are explanations of your questions.

  1. Abstract

In the abstract part, a great adjustment is made to clarify the research objectives and grouping information. Added the punch-related content and the results section.

  1. Introduction

We deleted the world on line 35 through line 40 what you said was not necessary. Added references of experimental studies related to punch. In addition, we explained the relationship between punch and joint.

  1. Research method

We described the process of subject recruitment and grouping in more detail and modified the table format as you suggested. We changed the group name from the control group to a beginner group.

  1. Data analysis

We still stick to the paired sample T-test, because the beginner group subjects were matched according to the age, height and weight of the master group subjects.

  1. Results

We indicated the statistical significance of each variable in Table 1 and Table 3.

  1. Discussion

We have made major revisions to the discussion.

Round 2

Reviewer 1 Report

I have no further comments to make.

Author Response

The paper has been revised and adjusted according to the recommendations of peer expert review. See attachment for details

Reviewer 2 Report

Thank you for your hard work on many revisions.

It is necessary to match the research purpose and conclusion of the title, abstract, and the research purpose of the introduction.

Is the author punching analysis the purpose of the study?

However, there is no mention of punching in the conclusion of the abstract.

And, I recommend that authors put punching in the title if they want to emphasize punching.

Author Response

(The authors gave the same response as above.)
